# MeshMosaic: Scaling Artist Mesh Generation via Local-to-Global Assembly

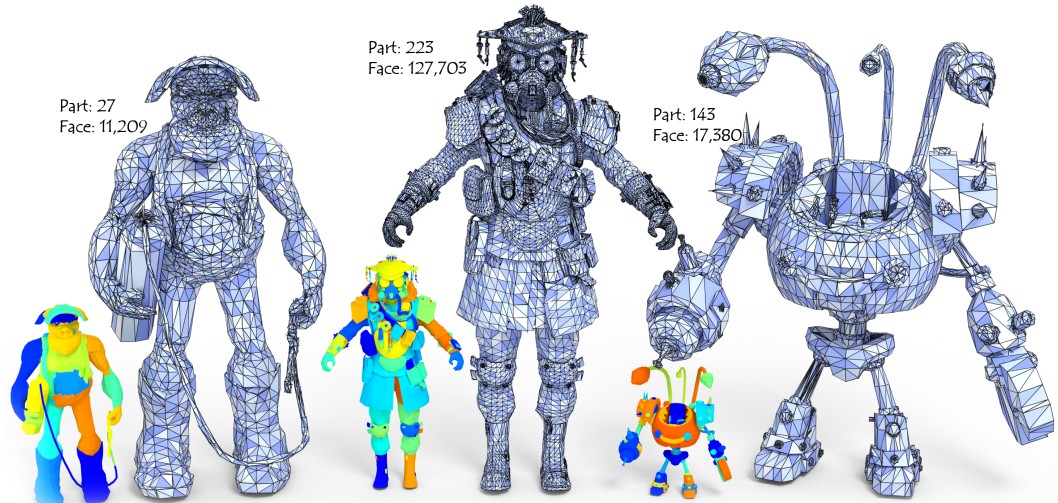

Figure 1: *MeshMosaic* empowers scaling up artist mesh generation to more than 100k triangles by assembling boundary-conditioned local patches into cohesive, high-resolution meshes. It delivers flexible support over mesh density and ensures the faithful retention of intricate design details. Faces are assigned random blue colors to better illustrate the mesh layout.

## Abstract

Scaling artist-designed meshes to high triangle numbers remains challenging for autoregressive generative models. Existing transformer-based methods suffer from long-sequence bottlenecks and limited quantization resolution, primarily due to the large number of tokens required and constrained quantization granularity. These issues prevent faithful reproduction of fine geometric details and structured density patterns. We introduce *MeshMosaic*, a novel local-to-global framework for artist mesh generation that scales to over 100K triangles—substantially surpassing prior methods, which typically handle only around 8K faces. *MeshMosaic* first segments shapes into patches, generating each patch autoregressively and leveraging shared boundary conditions to promote coherence, symmetry, and seamless connectivity between neighboring regions. This strategy enhances scalability to high-resolution meshes by quantizing patches individually, resulting in more symmetrical and organized mesh density and structure. Extensive experiments across multiple public datasets demonstrate that *MeshMosaic* significantly outperforms state-of-the-art methods in both geometric fidelity and user preference, supporting superior detail representation and practical mesh generation for real-world applications.

## 1 Introduction

Artist-designed triangular meshes are a cornerstone in film, gaming, AR/VR, and industrial design. High-quality artist meshes are central to computer graphics and 3D vision, distinguished by their stylized topology, directional flows, uneven triangle densities, sharp edges, and symmetry. Recent advances in 3D generation (Xiang et al., 2025; Li et al., 2024a; Liu et al., 2023; Long et al., 2024) and reconstruction (Wang et al., 2023; Huang et al., 2024; Xu et al., 2023; Hou et al., 2022) highlight

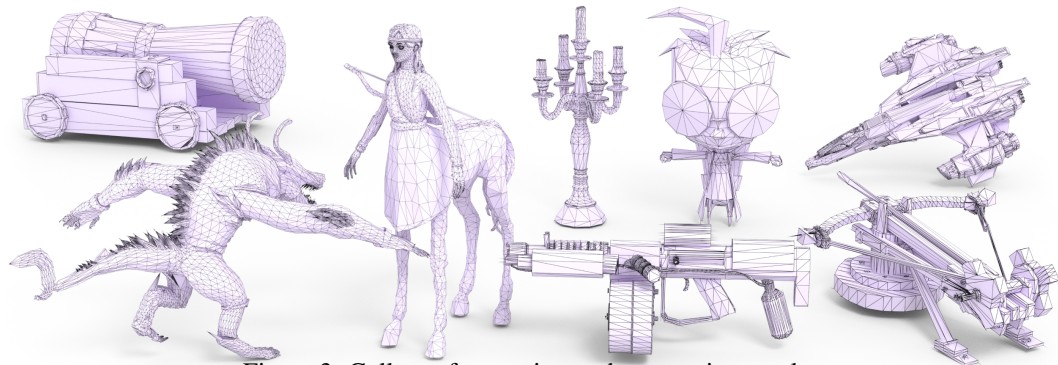

Figure 3: Gallery of our artist mesh generation results.

the limitations of classical meshing methods like Marching Cubes (Lorensen & Cline, 1998), which rely on uniform grids and produce redundant triangles, struggling with sharp features. Traditional meshing either yields uniform (Liu et al., 2009; Xu et al., 2024; Wang et al., 2025b; Dong et al., 2025b;a) or oversimplified (Chen et al., 2023; Garland & Heckbert, 1997) results; while anisotropic techniques (Zhong et al., 2014) better align to curvature, they still fall short in capturing the varying density and structure of artist meshes.

The rise of large language models (LLMs) (Zhao et al., 2023a) has inspired GPT-like architectures for mesh generation, such as MeshGPT (Siddiqui et al., 2024) and its successors (Chen et al., 2024a;c; Zhao et al., 2025; Hao et al., 2024; Tang et al., 2024b). Despite progress, these approaches struggle to scale up due to prohibitively **long token sequences** and **limited quantization resolution**, making it difficult to generate high-triangle meshes with fine detail. However, in practice, artist-designed meshes often require significantly higher resolutions to achieve the visual fidelity demanded in modern games and films. For example, production-quality character models or hero assets frequently contain upwards of 100K faces, far exceeding the capacities handled by current generative methods (Zhao et al., 2025; Weng

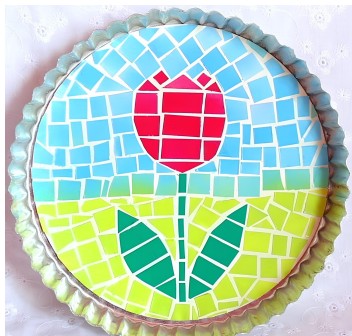

Figure 2: Mosaic Art (McPhee, 2025).

et al., 2025; Lionar et al., 2025). This substantial gap underscores the need for methods capable of generating high-triangle meshes that preserve the intricate details and structural coherence.

Inspired by the compositional principles of classical mosaic art (Fig. 2), we propose *MeshMosaic*, a novel local-to-global framework for scalable artist mesh generation. Mosaic artworks achieve global complexity and coherence by assembling intricate local tiles; in a similar spirit, *MeshMosaic* constructs a complete mesh by stitching together multiple locally generated patches. Unlike previous methods that attempt to model the entire mesh sequence, our framework divides the mesh into semantically meaningful patches, each autoregressively generated from a full-size point cloud with full-resolution quantization. To enable a compact yet faithful geometric representation, we employ shared boundary conditions and semantic segmentation, which address challenges related to boundary alignment and asymmetry. This patch-based strategy not only sidesteps the long-sequence bottleneck, but also effectively captures fine-grained geometric structures and global coherence, allowing for high-detail modeling within each patch while maintaining consistency across the entire mesh.

Experiments on multiple datasets show that *MeshMosaic* establishes new milestones in geometric fidelity and detail, also strongly preferred in user studies for artistry. Our approach supports stable generation of high-resolution meshes with over 100K triangles (see Fig. 1) and faithfully reproduces fine detail via per-patch quantization. See Fig. 3 for a gallery of our results.

Our key contributions are:

- We introduce a local-to-global autoregressive framework that decomposes meshes into patches, fundamentally overcoming the long-sequence bottleneck in mesh generation.

- We employ boundary-aware local quantization alongside semantic segmentation guidance, ensuring precise cross-patch alignment, symmetry preservation, global consistency, and stronger representation of intricate details.

- We achieve state-of-the-art results on multiple datasets, significantly outperforming baselines in fidelity and user preference.

## 2 RELATED WORKS

### 2.1 3D SHAPE GENERATION

Remarkable advances have been made in 3D shape generation, particularly with the adoption of signed distance field (SDF) representations, which offer notable accuracy and flexibility for modeling complex shapes. Despite such progress, these SDF-based methods often depend on the Marching Cubes algorithm (Lorensen & Cline, 1998) for mesh extraction, which can result in redundant triangles and consequently large file sizes—posing limitations for scalable deployment and real-time applications.

For instance, Wonder3D (Long et al., 2024) introduces a cross-domain diffusion framework for generating high-quality, multi-view textured 3D meshes from single images, achieving improved consistency and visual fidelity over previous approaches. CLAY (Zhang et al., 2024) expands the scope with a large-scale generative model that transforms text, images, and 3D-aware inputs into intricate geometry and material compositions, making robust 3D asset creation accessible to broad user bases. TRELLIS (Xiang et al., 2025) leverages structured occupancy fields to guide the formation of salient shape features, supporting high-precision modeling conditioned on text or image prompts. Hunyuan3D-2.5 (Lai et al., 2025) proposes a two-stage diffusion pipeline for crafting high-fidelity assets, combining powerful generative models with physically-based rendering for enhanced realism in both shape and texture. CraftsMan3D (Li et al., 2024b), evolves toward interactive 3D design by developing a native diffusion-based framework capable of producing meshes with regular topology and fine surface detail, while supporting user-driven refinements.

### 2.2 ARTIST MESH GENERATION

The quest for artist-quality mesh generation has inspired a new wave of models that focus on efficient topology and expressive geometry. MeshGPT (Siddiqui et al., 2024) pioneers autoregressive mesh synthesis through sequence-based modeling, employing quantized latent embeddings and transformer architectures to predict efficient triangulation and structural patterns reminiscent of hand-crafted meshes. Building on this idea, MeshAnything (Chen et al., 2024b) and MeshAnythingV2 (Chen et al., 2024c) offer advanced mesh generation using adjacent mesh tokenization, reducing token sequence lengths and enabling more complex, artist-grade meshes, with MeshAnythingV2 doubling the operational face limit.

MeshXL (Chen et al., 2024a) introduces the Neural Coordinate Field, which fuses explicit coordinate representation with implicit neural embeddings for more scalable, high-fidelity mesh modeling. EdgeRunner (Tang et al., 2024b) addresses past limitations of autoregressive mesh approaches by presenting an improved tokenization algorithm and compressing variable-length meshes into fixed-size latent vectors, yielding more diverse, generalizable, and higher-quality outputs. Meshtron (Hao et al., 2024) leverages a novel hourglass neural architecture with sliding window inference and robust sampling, achieving new levels of scalability and fidelity.

In addition, TreeMeshGPT (Lionar et al., 2025) introduces a tree sequencing method for triangle adjacency, dynamically growing mesh structures during autoregressive generation for improved training and mesh quality. iFlame (Wang et al., 2025a) balances efficiency and generative power by combining linear and full attention within an hourglass framework, augmented by caching for fast inference and training. Nautilus (Wang et al., 2025c) explores locality-aware autoencoding by leveraging manifold mesh properties, novel tokenization, and dual-stream conditioning, significantly enhancing scalability and structural consistency.

Compression-oriented approaches such as Blocked and Patchified Tokenization (BPT) (Weng et al., 2025) further reduce token sequence length, allowing detailed mesh synthesis with more faces. Building on BPT, DeepMesh (Zhao et al., 2025) integrates reinforcement learning for human preference alignment, supporting the generation of intricately detailed meshes with precise topology. In addition to autoregressive-based approaches, methods such as PolyDiff (Alliegro et al., 2023) and

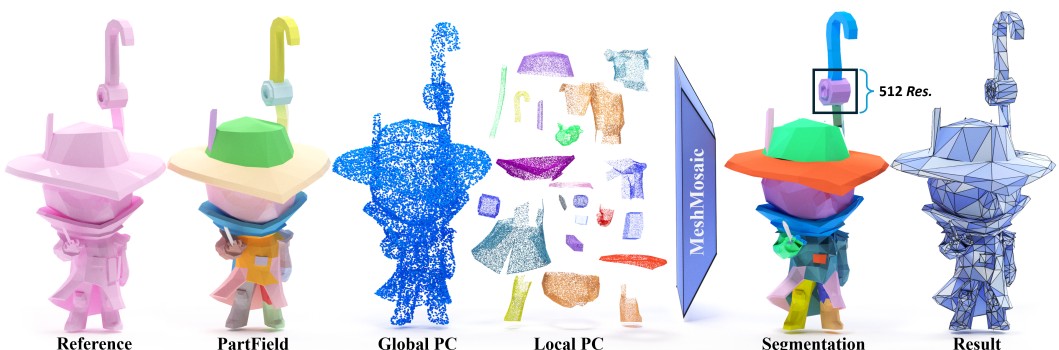

Figure 4: The pipeline of *MeshMosaic*. During inference, our method first applies PartField (Liu et al., 2025) to obtain semantic segmentation of the input shape. The input point cloud is then sampled according to the segmented patches and the original shape. Finally, our approach produces a clean, highly detailed mesh by assembling the generated patches.

PDT (Wang et al., 2025b) directly employ diffusion models to generate structured triangles or points from Gaussian noise.

## 2.3 PART-BASED SHAPE GENERATION

Part-based shape generation rests on the principle that decomposing objects into semantic parts furnishes rich priors for structure-aware reconstruction and controllable synthesis. Universal segmentation techniques have scaled part discovery across a wider range of data. Segment Any Mesh (Tang et al., 2024a) generalizes promptable segmentation to 3D meshes, supporting flexible and category-agnostic part extraction crucial for interactive and generative workflows. SAM3D (Yang et al., 2023) adapts this paradigm to large-scale 3D scenes, enabling multi-granular, prompt-driven segmentation. By distilling knowledge from SAM's multi-view segmentation results, SAMPart3D (Yang et al., 2024) further specializes in part segmentation for individual objects. More recently, PartField (Liu et al., 2025) represents shapes as continuous feature fields and trains a transformer-based feed-forward network with an ambiguity-agnostic contrastive loss, achieving efficient and high-quality open-world part segmentation. PartCrafter (Lin et al., 2025) jointly creates multiple semantically distinct parts from a single image, enabling end-to-end part-aware 3D mesh synthesis with global coherence and fine-grained detail.

Human modelers typically create models based on their understanding of component-based structures (Lin et al., 2020), and thus part-based generation is a problem of significant importance. Part123 (Liu et al., 2024) illustrates this by reconstructing 3D shapes from single images while predicting semantic parts and their spatial arrangement. ComboStoc (Xu et al., 2024) introduces combinatorial stochasticity into diffusion by jointly sampling discrete structural decisions (such as part templates or multiplicity) with continuous geometry. These segmentation frameworks underpin part-based shape generation by providing scalable, promptable part vocabularies and supporting interactive conditioning and evaluation at the part level.

## 3 METHOD

### 3.1 OVERVIEW

Given a 3D reference shape, our target is to generate an artistic triangle mesh from it (see Fig. 4). *MeshMosaic* decomposes this task into a patch-by-patch generation process, allowing the generation of more triangles to carve details. First, we segment the shape into multiple different patches and determine their sequential order (Sec. 3.2). Next, we introduce an innovative approach that incorporates boundary and global context as conditioning information for each individual patch (Sec. 3.3). Finally, we present the training methodology for this framework (Sec. 3.4).

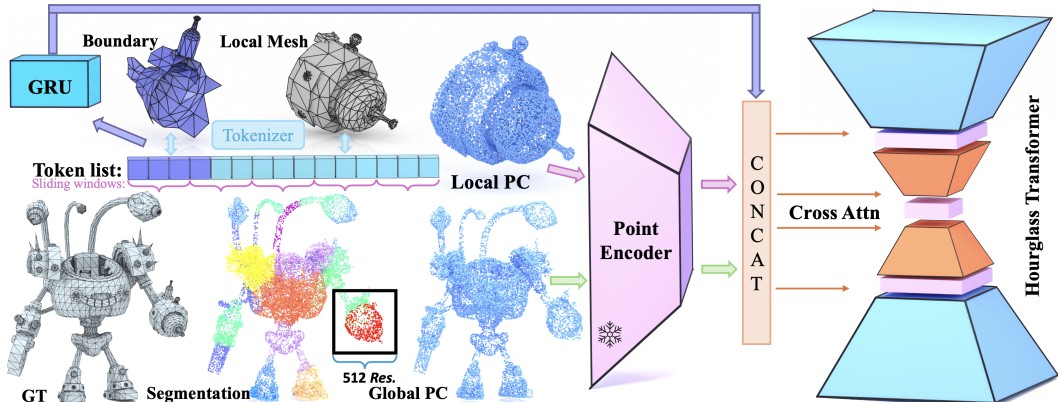

Figure 6: The workflow of *MeshMosaic* for generating a single patch. Both global and local point cloud features are extracted by a locked Michelangelo (Zhao et al., 2023b) encoder. For each patch, the nearest boundary mesh is identified, tokenized, and concatenated before the target mesh token sequence. The GRU network encodes boundary tokens, which are then combined with global and local features and fed into an autoregressive hourglass transformer for mesh generation.

## 3.2 LOCAL-TO-GLOBAL MESH GENERATION

**Semantic Patch Segmentation.** Autoregressively generating the complete shape directly can be problematic, since such networks must handle long token sequences and may struggle to represent fine geometric details due to limited quantization resolution. By decomposing the shape into multiple patches and generating them sequentially, these issues are largely mitigated, and each patch maintains fine granularity while keeping network input manageable.

We use PartField (Liu et al., 2025) for semantic segmentation at inference time (see Fig. 4), which embeds semantic structure and produces well-aligned boundaries, often guided by curvature flow to enhance realism and make future edits easier.

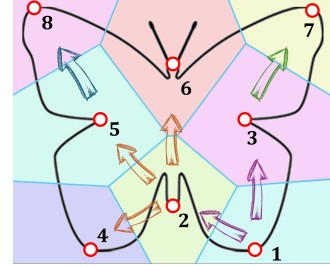

**Sorting patches.** Then, we will generate the whole mesh in a part-by-part manner, which requires us to determine a generation order. Thus, the patch generation is carried out in breadth-first search (BFS) order, beginning from the spatially lowest patch and then proceeding to adjacent patches. Sequential generation with the autoregressive model ultimately yields the final mesh assembly. Fig. 5 shows a 2D illustration with eight patches, where the black line demonstrates the

Figure 5: 2D illustration of patches with BFS order.

mesh surface. Then, for each patch, we adopt the following structure to generate the triangle meshes.

## 3.3 GENERATING SINGLE PATCH

We next concentrate on generating individual patches. Simply applying the same network architecture on every patch without considering connection relationship risks continuity issues, such as broken boundaries, irregular density, or lost symmetry. Fig. 6 illustrates our dedicated architecture for generating individual patches. The following paragraphs elaborate on our solutions to these specific challenges.

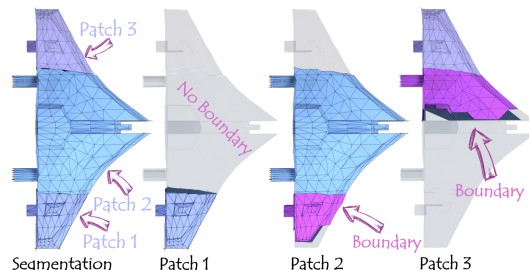

Figure 7: Example of boundary condition.

**Constructing Boundary Condition.** When generating triangles for a specific patch, we will use the triangles from existing generated patches as boundary conditions. This essentially enables the smooth connection between different patches. Specifically, we introduce an efficient boundary conditioning mechanism: the token sequence from earlier patches is fed as context to subsequent patches. To avoid inefficiency and information dilution from excessively long token sequences, for each patch, we select only 512 spatially nearest triangles from prior patches. These are tokenized, passed through

a Gate Recurrent Unit (Cho et al., 2014) (GRU) network, and the resulting embedding conditions the transformer network (see Fig. 6, blue arrows). Fig. 7 shows an example of the plane shape. The whole shape was segmented into three patches. Each patch gets the boundary information from the previous patches, following the BFS order. For the very first patch, where no previous boundaries exist, we supply a placeholder token sequence consisting entirely of terminator tokens to the GRU network, establishing a neutral starting context for the generation process.

**Injecting Boundary Condition.**   Given the encoded boundary triangle information, we then inject such information into the generation process of the current patch. We concatenate the boundary condition tokens to the beginning of the target patch's token sequence. This approach allows the model to leverage self-attention mechanisms over both the boundary and the patch-specific tokens (see the colored tokens in Fig. 6). By integrating boundary information directly into the patch generation context, we ensure that the triangles along the shared boundaries naturally extend and blend into neighboring patches. Ablation studies demonstrating the effect of these boundary conditions are discussed in Appendix Sec. A.2.

**Local-to-Global Conditioning.**   While boundary conditioning enforces local continuity, we enhance global coherence via local-to-global point cloud features (middle of Fig. 6). During training and inference, our autoregressive model is conditioned on representations from both the current patch point clouds and the full shape point clouds. Both point clouds are encoded with a frozen Michelangelo (Zhao et al., 2023b) encoder. The extracted global and local features are concatenated with GRU boundary features and provided to the transformer as final condition input.

**Local Quantization.**   Given both the boundary conditions and local-to-global information, we generate the triangles locally using a local quantization. Unlike previous approaches, such as DeepMesh (Zhao et al., 2025), which apply a uniform quantization of $512^3$ resolution to the entire mesh, our method independently scales each patch to $[0, 1]$ and quantizes it at $512^3$ resolution. This local quantization approach enables a higher effective merged resolution, allowing for the preservation and recovery of richer geometric details. As shown in Fig. 4 and Fig. 6, our pipeline first segments the shape into patches, with each patch quantized independently to $512^3$ resolution and provided with 16,384 sampled points as input. In contrast to baseline methods, which quantize the full shape and use only a single set of 16,384 point cloud samples, our framework assembles meshes from individually quantized patches, each paired with its respective sampled points. This strategy offers the dual benefits of higher overall shape resolution and a greater abundance of conditional information for the network to leverage during generation.

**Gluing local patches.**   It should be noted that local quantization may introduce minor positional displacements for each patch, which can lead to discontinuities along patch seams if not properly addressed. To ensure seamless integration, we compute the displacement between the position of boundary condition faces referenced by the current patch and their corresponding original quantized positions in the previously assembled patches. The entire current patch is then translated according to this computed displacement, aligning it precisely with previously assembled patches. This compensatory adjustment guarantees smooth boundaries and continuity across the entire mesh, enabling high-fidelity splicing between patches, producing a unified and detailed final mesh.

## 3.4   TRAINING STRATEGY

Our training begins by segmenting the input mesh and extracting boundary information for autoregressive conditioning. Semantic segmentation is omitted during training because it is relatively time-consuming and reduces diversity. Instead, our approach utilizes random segmentation, which promotes better network diversity and scalability.

Each mesh is partitioned into patches adaptively in training. Given an input mesh $\mathcal{M}$ with $\mathcal{N}_p$ vertices and $\mathcal{N}_f$ facets, the number of patches is set to $\mathcal{N}_{seg} = \frac{\mathcal{N}_f}{2000} \times \lambda_{rand}$, where $\lambda_{rand}$ is randomly sampled from $[0.5, 2.5]$ to encourage diversity. The denominator ensures that each patch, after tokenization, yields a sequence length close to the window size (9K) for efficient training.

We apply farthest point sampling (Moenning & Dodgson, 2003) to select $\mathcal{N}_{seg}$ points as cluster centers. Voronoi decomposition (Aurenhammer, 1991) partitions the mesh into patches based on

Table 1: Quantitative comparison on ShapeNet (Chang et al., 2015), Thingi10K (Zhou & Jacobson, 2016) and Objaverse (Deitke et al., 2023b) datasets. The **best** scores are emphasized in bold with underlining, while the **second best** scores are highlighted only in bold.

| Dataset | Method | HD ↓ | $CD_{L1}$ ↓ | $CD_{L2}$ $(\times 10^3)$ ↓ | NC ↑ | F1 ↑ | ECD ↓ | EF1 ↑ |
|---------|--------|------|-------------|------------------------------|------|------|-------|-------|
| ShapeNet | MeshAnythingV2 | 0.078 | 0.009 | 0.640 | 0.911 | 0.652 | 0.055 | 0.130 |
| | BPT | **0.017** | **0.003** | **0.012** | 0.962 | **0.875** | **0.040** | **0.159** |
| | TreeMeshGPT | 0.161 | 0.034 | 5.430 | 0.841 | 0.556 | 0.089 | 0.100 |
| | DeepMesh | 0.037 | 0.004 | 0.060 | **0.967** | 0.791 | 0.056 | 0.177 |
| | Ours | **0.037** | 0.003 | 0.019 | **0.973** | **0.929** | 0.052 | **0.211** |
| Thingi10K | MeshAnythingV2 | 0.167 | 0.021 | **2.492** | 0.842 | 0.358 | 0.036 | 0.110 |
| | BPT | **0.157** | 0.035 | 7.771 | **0.875** | **0.496** | 0.051 | **0.179** |
| | TreeMeshGPT | 0.233 | 0.060 | 18.086 | 0.788 | 0.387 | 0.057 | 0.161 |
| | DeepMesh | 0.165 | **0.026** | 3.331 | 0.853 | 0.321 | **0.031** | 0.137 |
| | Ours | **0.051** | **0.004** | **0.052** | **0.942** | **0.746** | **0.017** | **0.271** |
| Objaverse | MeshAnythingV2 | 0.118 | 0.015 | 1.213 | 0.859 | 0.430 | 0.021 | 0.115 |
| | BPT | 0.151 | 0.034 | 7.016 | 0.846 | **0.502** | 0.027 | 0.164 |
| | TreeMeshGPT | 0.237 | 0.057 | 10.507 | 0.784 | 0.308 | 0.067 | 0.072 |
| | DeepMesh | **0.111** | **0.016** | **1.712** | **0.866** | 0.471 | **0.021** | **0.168** |
| | Ours | **0.072** | **0.007** | **0.387** | **0.919** | **0.785** | **0.006** | **0.348** |

these centers, with bisecting planes separating triangle regions. Clusters are ordered by breadth-first search, starting from the lowest center, to retrieve boundary information sequentially.

We also curate a subset of meshes with high-quality connected component annotations, using each component directly as a patch (with breadth-first ordering, as above). This subset supports tasks requiring more regular, consistent patch segmentation and enables training for semantic reasoning. We provide detailed training dataset analysis is in Appendix Sec. A.1.

## 4 EXPERIMENTS

### 4.1 IMPLEMENTATION

Our implementation builds upon the released $0.5B$ parameter DeepMesh (Zhao et al., 2025) model, which serves as the base for fine-tuning our approach. We introduce and progressively fuse new boundary conditions and global point cloud features into the architecture, connecting the GRU boundary encoder and global feature input using zero-initialized linear layers. Local point cloud features are mapped directly onto the original input cloud.

The curated dataset consists of 310K meshes, including approximately 90K with connected component information. The distribution of face count are visualized in Fig. 8.

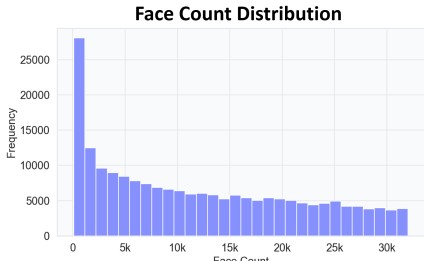

Figure 8: Distribution of face count in our dataset.

Please check Appendix Sec. A.1 for the data preprocess and more analysis. Training is conducted for seven days on a cluster of 32 NVIDIA H20 96GB GPUs, using a cosine learning rate scheduler that decays from $1 \times 10^{-4}$ to $1 \times 10^{-5}$. Token window sizes for truncated windows follow DeepMesh's setting ($9K$, with a $50\%$ overlap). We employ KV-caching in both training and inference and adopt probabilistic sampling (temperature $0.5$) to ensure stable mesh generation.

### 4.2 COMPARISONS

To thoroughly assess the effectiveness of our proposed method, we perform comparative experiments with four publicly available state-of-the-art mesh generation methods: MeshAnythingV2 (Chen et al., 2024c), BPT (Weng et al., 2025), TreeMeshGPT (Lionar et al., 2025), and DeepMesh (Zhao et al., 2025). The comparison includes both quantitative measurements and qualitative visualization of results. We randomly select 100 samples from each of the ShapeNet (Chang et al., 2015), Thingi10K (Zhou & Jacobson, 2016), and Objaverse (Deitke et al., 2023b) datasets for all experiments.

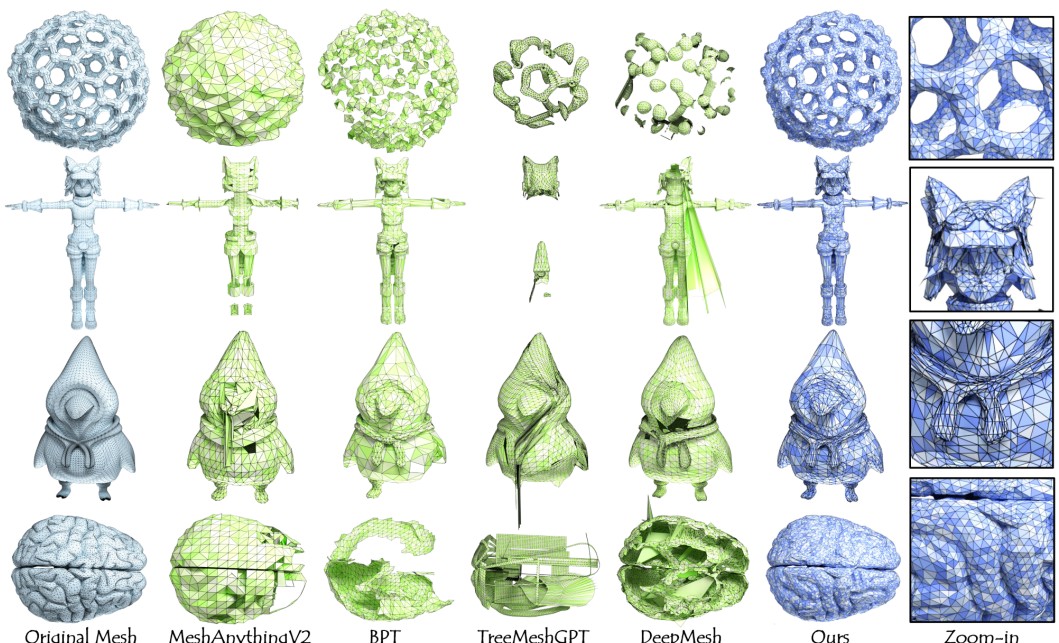

Original Mesh  MeshAnythingV2  BPT  TreeMeshGPT  DeepMesh  Ours  Zoom-in

Figure 9: Visual comparison of *MeshMosaic* with SOTA methods. The first row shows the input shapes; the last row highlights detailed close-ups of meshes generated by our method. Faces are randomly colored to highlight the mesh layout.

**Geometric Metrics.** As a mesh generation framework, faithfully preserving both the overall shape and fine-grained details of the original object is paramount; notable deviations from the reference geometry are unacceptable. To quantitatively measure the fidelity between the generated mesh and the ground-truth shape, we utilize four widely adopted evaluation metrics: *Hausdorff Distance* (HD), *Chamfer Distance* (CD), *Normal Consistency* (NC), and *F-score* (F1). Following CWF (Xu et al., 2024), we additionally incorporate *Edge Chamfer Distance* (ECD) and *Edge F-score* (EF1), as introduced by NMC (Chen & Zhang, 2021), to specifically assess the preservation of sharp features.

As summarized in Tab. 1, our proposed method consistently surpasses all baseline approaches across almost all datasets and evaluation metrics. *MeshMosaic* not only excels in geometric accuracy but also demonstrates significant improvements in retaining intricate features and ensuring overall mesh quality. These comprehensive results underscore the effectiveness and robustness of our approach for high-fidelity mesh generation.

Qualitative comparisons in Fig. 9 further illustrate that our method produces meshes with higher fidelity and finer detail. By contrast, MeshAnythingV2 (Chen et al., 2024c) and BPT (Weng et al., 2025) yield meshes of relatively lower quality and resolution, resulting in the loss of high-frequency details. TreeMeshGPT (Lionar et al., 2025) and

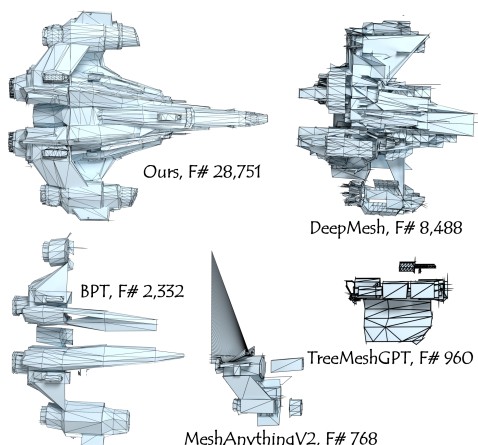

Figure 10: Comparison on triangle numbers.

DeepMesh (Zhao et al., 2025), while capable of generating denser meshes, utilize global one-shot autoregressive mechanisms and consequently struggle to capture complex geometries, such as those evident in the first and last examples. In contrast, our approach leverages a local-to-global prior generation strategy, which not only ensures structural correctness but also enhances the representation of subtle features. Moreover, although datasets like ShapeNet (Chang et al., 2015), Thingi10k (Zhou & Jacobson, 2016), and Objaverse (Deitke et al., 2023b) exhibit varying complexity, our method consistently outperforms competing methods across all of them.

We present a more compelling example in Fig. 10. For a complex fighter jet model, our method successfully reconstructs intricate details using nearly 30K triangles whereas other approaches

Table 2: User study with SOAT methods aggregated from 27 professional participants in four categories: *Neatness*, *Artistry*, *Similarity to Ground Truth*, and *Detail Recovery*. The **best** scores are emphasized in bold with underlining, while the **second best** scores are highlighted only in bold.

| Method | Neatness ↑ | Artistry ↑ | Similarity to GT ↑ | Detail Recovery ↑ |
|---|---|---|---|---|
| MeshAnythingV2 | 0.864 | 0.780 | 0.612 | 0.628 |
| BPT | **1.040** | **0.932** | **1.072** | **1.084** |
| TreeMeshGPT | 0.696 | 0.684 | 0.600 | 0.512 |
| DeepMesh | 0.712 | 0.808 | 0.772 | 0.848 |
| Ours | **2.780** | **2.785** | **2.912** | **2.912** |

struggle with such highly complex shapes, typically yielding only a few hundred to a few thousand triangles. This noticeable gap demonstrates the superior detail recovery and scalability of our method with substantially higher triangle counts.

**User Study.** Beyond reconstruction accuracy, it is crucial that generated meshes are artist-quality with sparse, neat, visually compelling, and easy to edit meshes. To assess this, we conducted a user study, sampling 10 models from test datasets. Twenty-seven professional users with expertise in computer graphics or 3D modeling anonymously rated five methods on four criteria: *Neatness*, *Artistry*, *Similarity to Ground Truth*, *Detail Recovery*. Scores were assigned for the top three methods in each category (3, 2, and 1 points, respectively; 0 for others). The final scores are summarized in Tab. 2. *MeshMosaic* achieved the highest ratings in all categories, reflecting its superior aesthetic and structural quality. Competing methods scored lower due to issues with mesh stability, single-pass autoregressive models often stall or fail for long, complex meshes, yielding incomplete outputs. BPT (Weng et al., 2025) ranked second in the user study, reflecting similar trends in reconstruction metrics in Tab. 1. Although BPT (Weng et al., 2025) tends to produce more stable outputs, its overall mesh quality is comparatively lower and struggles to preserve fine details. This is further evidenced by its performance in the ECD and EF1 metrics: while BPT (Weng et al., 2025) delivers satisfactory results for relatively simple shapes, such as those in ShapeNet (Chang et al., 2015). Its scores decline markedly with increasing shape complexity (Thingi10K (Zhou & Jacobson, 2016) and Objaverse (Deitke et al., 2023b) datasets).

## 5 DISCUSSION

**Conclusion.** We present *MeshMosaic*, a boundary-conditioned local-to-global autoregressive framework that decomposes meshes into compact patches and assembles them coherently. This design fundamentally removes the long-sequence bottleneck and enables higher-resolution quantization, scaling generation to over 100K triangles while preserving fine-grained geometric detail. On ShapeNet (Chang et al., 2015), Thingi10K (Zhou & Jacobson, 2016), and Objaverse (Deitke et al., 2023b) dataset, *MeshMosaic* achieves state-of-the-art fidelity and perceptual quality, consistently surpassing the baselines. Beyond meshes, it offers a general paradigm for scaling autoregressive generation of structured 3D data via patch-level modeling.

**More Discussions.** We also present additional discussions and comprehensive ablation studies in Appendix Sec. A.2. Including ablations for the various proposed conditions, more detailed comparisons, analyses of different segmentation inputs, evaluations of text and image inputs, running time assessments, and diversity metrics, among others.

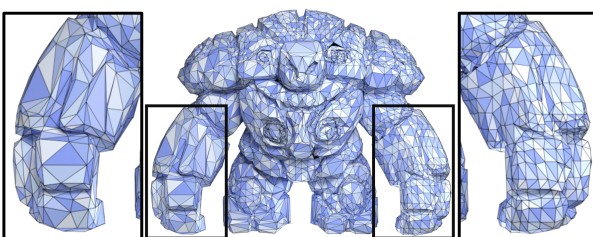

Figure 11: Symmetry limitation.

**Limitations and future works.** While *MeshMosaic* enforces local coherence, boundary conditioning remains primarily local and may leave distant symmetric parts weakly coupled. As illustrated in Fig. 11, the two arms exhibit mild asymmetry despite reasonable connectivity and density. When stronger symmetry is required, this could be alleviated by incorporating global perception mechanisms to couple distant parts. Beyond symmetry, future work may explore cross-patch refinement, multimodal conditioning, and adaptive quantization to further enhance resolution and editability.

**Reproducibility statement.** *MeshMosaic* is developed entirely on top of the open-source DeepMesh (Zhao et al., 2025) framework and fine-tuned using publicly available checkpoints. To reproduce *MeshMosaic* results, users simply need to incorporate the global awareness and boundary awareness modules into the existing DeepMesh (Zhao et al., 2025) code. We will be releasing our code and checkpoints in the near future.

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

# A APPENDIX

## A.1 DATA PREPROCESSING

Training datasets are drawn from Objaverse-XL (Deitke et al., 2023a) and other licensed datasets. To enhance data quality, we implemented several filtering procedures. Only meshes with $[500, 32000]$ faces are retained, excluding those with excessively low or high token lengths. Meshes are subsequently cleaned and optimized using PyMeshlab (Muntoni & Cignoni, 2021): duplicate vertices/faces removed, closely spaced or overlapping vertices merged, non-manifold elements and edges eliminated.

And we computed a point-to-face ratio for each model:

$$\mathbf{\Phi}_{\mathbf{p/f}} = \frac{\mathcal{N}_p}{\mathcal{N}_f} \tag{1}$$

Meshes with $\mathbf{\Phi}_{\mathbf{p/f}} > 0.8$ are filtered out to exclude objects with too many open boundaries. For robustness, we augment data via random rotations with three axes and uniform scaling within $[0.9, 1.0]$.

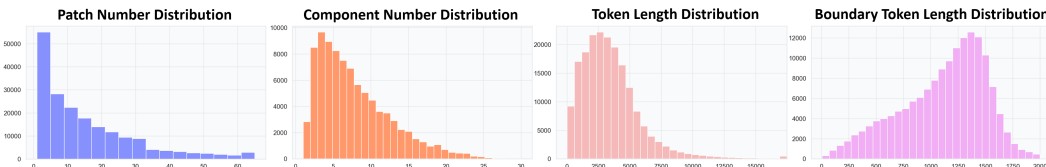

Figure 12: Dataset statistics: from left to right (1) distribution of number of patches per mesh; (2) number of connected components for partially connected components; (3) token length per training patch; (4) token length of boundary condition sequences.

Fig. 12 provides a comprehensive analysis of our dataset. Moving from left to right, the first graph illustrates the distribution of the number of patches generated through random segmentation. Most simple shapes are divided into fewer than ten patches, whereas a small number of highly complex cases yield over 60 patches. Although our training set contains no more than sixty splits per instance, our method can handle inference tasks involving hundreds of patches during testing. This highlights the strong generalization capability of our method (as shown in Fig. 1).

Next, we report statistics on the number of connected components for samples with native splits as previously noted. Most samples containing fewer than ten patches, similar to the distribution observed from random splits.

Facilitated by our local-to-global architecture, the required token length for each training or inference phase is significantly diminished. We present the distribution of token lengths for all split patches. The vast majority contain fewer than 6,000 tokens, with the longest sequence not exceeding 20,000 tokens. This approach allows us to break down challenging problems into several manageable subproblems, each of which can be solved independently. Lastly, we report that boundary condition tokens are much shorter than full tokens, with all lengths falling below 2,000 tokens.

## A.2 DISCUSSION AND ABLATION

**Ablation Study.** We perform an extensive ablation study to systematically examine the roles of boundary conditions and global point cloud features within our mesh generation architecture. This analysis provides critical insights into how each conditioning mechanism contributes to the fidelity and coherence of generated meshes.

As illustrated in Fig. 13, we analyze three distinct ablated configurations: (1) **Ours w/o GPC**, in which the global point cloud conditioning feature is entirely removed; (2) **Ours w/o BD**, where the GRU network responsible for boundary condition encoding is omitted; and (3) **Ours w/o SA**, which disables the concatenation of boundary tokens for self-attention within the network.

To ensure a thorough assessment, ablation experiments are conducted under two regimes. The first regime (top row in Fig. 13) involves a controlled overfitting scenario, where the network is trained

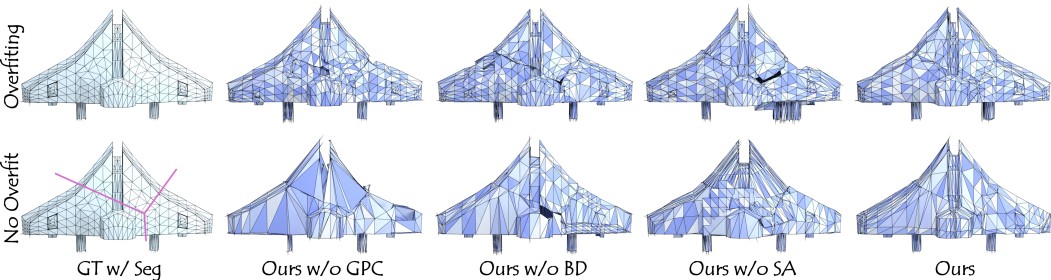

Figure 13: Ablation for boundary condition and global point cloud.

exclusively on a single airplane mesh for 20 epochs with a batch size of 8; segmentation boundaries are randomized at each iteration to probe the model's adaptability and generalization. The second regime (bottom row) evaluates the network after comprehensive training on our entire dataset, thereby measuring its capability across diverse object geometries.

For consistent comparison and clear visualization, all results in Fig. 13 utilize an identical segmentation scheme, indicated by the purple dividing line, which partitions each shape into three patches at inference time.

When global point cloud information is omitted (**Ours w/o GPC**), the network demonstrates reasonable performance in the overfitted regime, as it only needs to reconstruct a single shape. However, in the full dataset setting, the absence of global context leads to significant errors—most notably, the right portion of the mesh exhibits pronounced deformation and collapse, revealing the necessity of global information for guiding overall shape reconstruction. When the GRU-based boundary encoding is eliminated (**Ours w/o BD**), visible cracks emerge along the seams in both regimes. In addition, the absence of boundary communication induces substantial mesh density asymmetry in the full dataset setting, with adjacent patches developing inconsistencies. This reflects the model's inability to properly propagate local information between neighboring patches. Disabling the concatenation of boundary tokens for self-attention (**Ours w/o SA**) again results in prominent seam artifacts, and in the full dataset scenario, produces overlapping, self-intersecting patches. The lack of explicit constraints leads to independent patch generation, which ultimately causes geometric inconsistencies and structural artifacts.

In contrast, the full model employing both boundary and global conditioning produces meshes that are complete, uniform, and visually coherent, with mesh density and topology smoothly balanced across all patches. This clearly demonstrates the effectiveness of our proposed integration of local and global context, and highlights the importance of both conditional mechanisms for high-fidelity mesh generation.

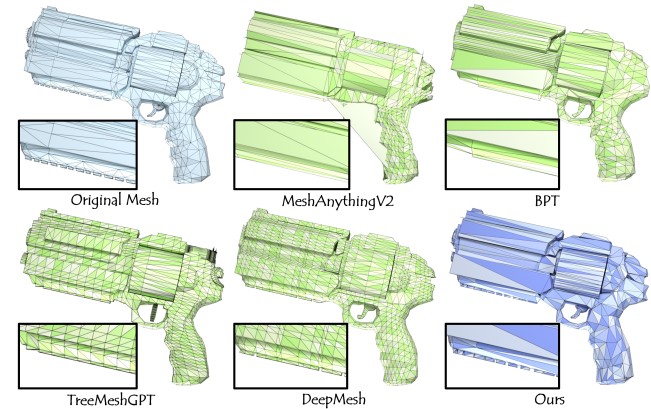

Figure 14: Detail recovery comparison.

**Detail Eecovery.** Thanks to our local-to-global sequential mesh generation strategy, our method significantly surpasses previous approaches in detail preservation. Unlike other methods that rely on a single quantized resolution for the entire model, our approach assigns an independent $512^3$ resolution to each patch. As illustrated in Fig. 14, our method is uniquely capable of recovering the original edge details of the pistol, whereas competing methods either fail to capture these features or merge them into indistinct blocks.

**Segmentation Input.** Although our approach is primarily designed to operate under a segmented training and inference regime, it nevertheless retains the flexibility to infer simple shapes without

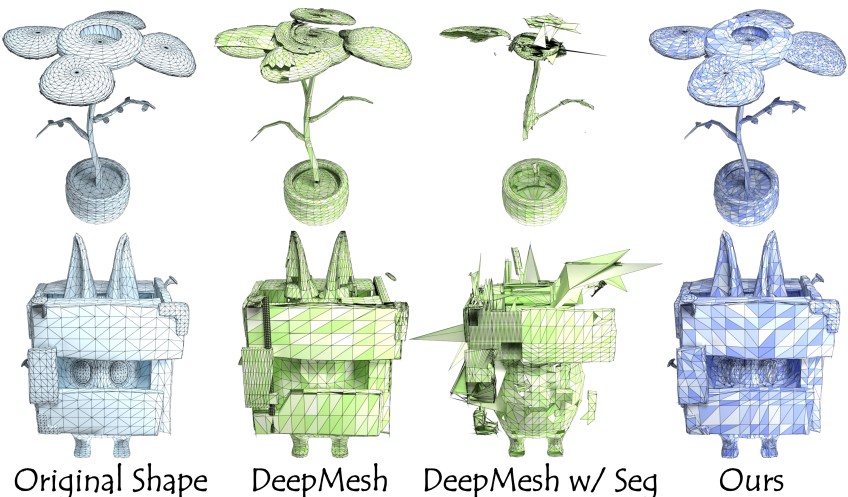

Original Shape    DeepMesh    DeepMesh w/ Seg    Ours

Figure 17: Our method without segmentation.

explicit segmentation. As demonstrated in Fig. 15, both our method and DeepMesh (Zhao et al., 2025) are capable of reconstructing a torus model in the absence of any segmentation.

Further analysis of segmentation strategies is shown in Fig. 16. In the middle example, reconstruction is performed using random segmentation. While the overall shape and fine details can still be recovered, the absence of semantic segmentation often results in patch boundaries that traverse flat or non-essential regions, introducing visual clutter and irregularity into the mesh appearance. By employing PartField (Liu et al., 2025) for semantic guidance, our method achieves noticeably cleaner and more coherent mesh boundaries, significantly enhancing the aesthetic quality without compromising reconstruction fidelity.

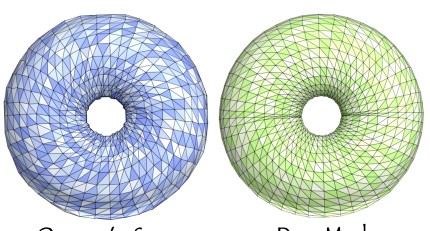

Ours w/o Seg    DeepMesh

Figure 15: Inference without segmentation.

**Comparison with DeepMesh.** Directly scaling DeepMesh (Zhao et al., 2025) to our local-to-global setting is non-trivial. To further demonstrate the benefits of our local-to-global framework and the importance of our boundary condition method, we perform an ablation study that directly compares it with DeepMesh (Zhao et al., 2025). As depicted in Fig. 17, we assess two distinct inference settings for DeepMesh: the first utilizes the entire shape without segmentation, while the second processes each segmented patch individually and subsequently assembles them to form the complete object.

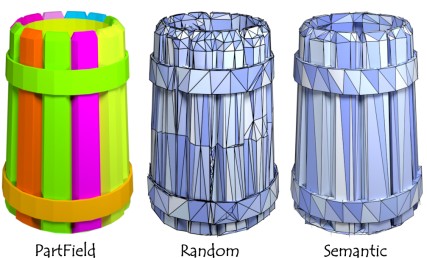

PartField    Random    Semantic

Figure 16: Comparison of random and semantic segmentation.

In the first scenario, where DeepMesh generates the mesh from an unsegmented input, it succeeds in producing reasonable global geometry. However, the quality of reconstructed fine details—such as the eye region in the second example is noticeably lacking. This demonstrates DeepMesh's limitations when handling intricate local features under a global, one-shot autoregressive scheme.

In the second scenario, we input our segmented data into DeepMesh, allowing it to process each patch independently. Local mesh resolution is indeed improved due to smaller region-specific quantization. Nonetheless, the absence of key contextual mechanisms: explicit boundary conditions and global shape information, leads to significant artifacts. The resulting meshes exhibit poor coherence across patch boundaries, with misaligned regions and inconsistent topology.

By contrast, our local-to-global strategy explicitly conditions each segment on both boundary and global cues, enabling seamless integration and faithful reconstruction of complex features throughout the mesh. This comparative analysis clearly highlights the expressive superiority and practical robustness of our method, especially in scenarios that demand high-resolution details and structurally consistent results.

**Runtime.** We developed our method on the DeepMesh (Zhao et al., 2025) codebase, thereby ensuring a comparable runtime environment and a rigorous basis for performance assessment. Tab. 3 details the training and inference efficiency of DeepMesh versus our proposed framework, including variants with specific ablations such as the boundary condition encoding (**BD**) and global point cloud encoding (**GPC**).

Table 3: Comparison of runtime performance between DeepMesh and our method variants. The table reports the training time per window (9K tokens) and the inference time per token in seconds.

|       | DeepMesh | Ours w/o BD | Ours w/o GPC | Ours  |
|-------|----------|-------------|--------------|-------|
| Train | 0.451    | 0.531       | 0.558        | 0.633 |
| Infer | 0.025    | 0.024       | 0.024        | 0.024 |

By incorporating GRU-based boundary condition encoding and a global point cloud conditioning module into our pipeline, we necessarily introduce additional computational operations during the training stage. This enhancement results in a moderate increase in training time relative to the original DeepMesh (Zhao et al., 2025) implementation.

However, our approach leverages the KV-Cache technique to substantially accelerate inference. All conditional features from global and boundary sources are preprocessed once at the beginning of the inference stage and then cached for subsequent decoding steps. This enables our method to maintain an average per-token inference time that is nearly equivalent to DeepMesh, regardless of ablation configuration, thereby ensuring strong deployment efficiency and scalability.

It is important to note that the overall inference time for any given model depends linearly on the total number of tokens generated. When the number of tokens is held constant, our method achieves inference performance on par with DeepMesh. Crucially, the local-to-global segmentation strategy of our method allows for the generation of meshes containing substantially more polygons, thereby supporting finer geometric detail and more complex structures. This increase in expressive capability is reflected in proportionally longer token sequences, resulting in a higher absolute inference time for such rich meshes. Nevertheless, the per-token efficiency of our method remains high, and any increase in total inference time is attributable to the practical need for representing more detailed and high-resolution outputs. However, although we use KV-cache to accelerate inference, and inference time is only about 0.024 seconds per token, inference on a very complex mesh can still take a very long time. For example, as shown by the mesh in the middle of Fig. 1, when the number of faces exceeds 100K, inference typically requires several hours to complete. This remains far from meeting the efficiency demands of industrial applications.

**Text and Image-Conditioned Generation.** Generating 3D shapes from text or image inputs has become a prominent direction in computer graphics and generative modeling, with recent advances delivering impressive results in open-domain shape synthesis. However, many contemporary techniques, particularly those relying on Signed Distance Functions (SDF), produce meshes by converting dense volumetric grids via algorithms like marching cubes. This process often results in excessive and redundant triangles, leading to overly complex meshes that are inefficient for practical applications in animation, rendering, or interactive editing.

In Fig. 18, we showcase examples where state-of-the-art SDF-based methods, such as CLAY (Zhang et al., 2024), generate initial 3D geometry from either textual prompts or image inputs. We then refine these preliminary outputs using *MeshMosaic*, producing artist-quality meshes that retain rich geometric details while optimizing triangle utilization. Compared to the raw outputs from CLAY, meshes processed by our framework exhibit cleaner topology, enhanced visual fidelity, and improved efficiency, making them far better suited for real-world downstream tasks. These results highlight the effectiveness of our method for transforming dense generative outputs into structured, high-quality assets tailored for professional use.

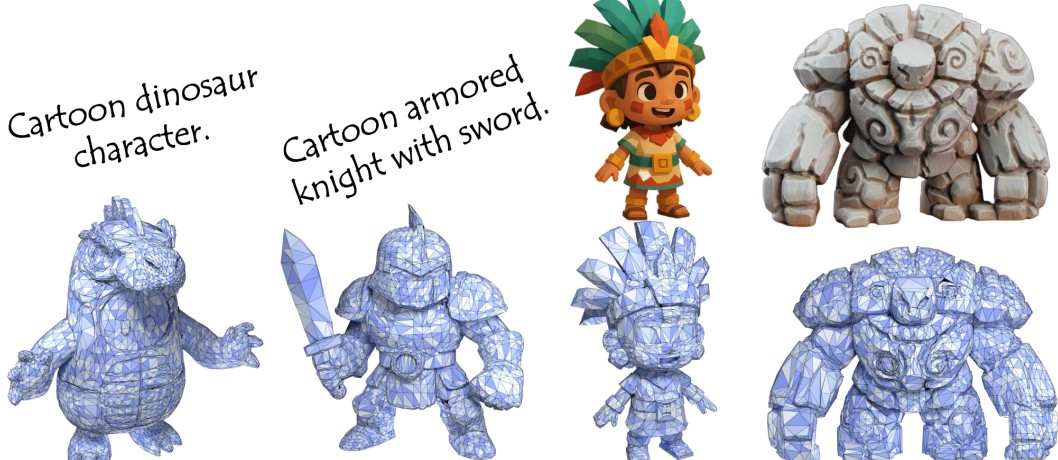

Figure 18: Results generated by *MeshMosaic* using text prompts (left) or image inputs (right). Initial 3D shapes are created using CLAY (Zhang et al., 2024) and enhanced by our approach.

**Diversity Generation.** We further illustrate the versatility and diversity of mesh outputs produced by *MeshMosaic*. As depicted in Fig. 19, our framework is capable of generating a broad spectrum of meshes even when provided with an identical point cloud input. This demonstrates

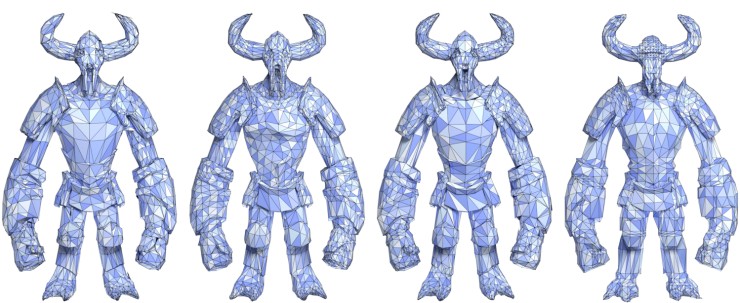

Figure 19: Diversity of our generation results.

the network's intrinsic capacity for structural variation and contextual adaptation. For example, in the minotaur warrior scenario, our method synthesizes markedly distinct mesh representations for different anatomical and accessory regions—including chest armor, shoulder plates, arms, and head. Each of these regions features unique geometric patterns and connectivity details, clearly reflecting localized artistic interpretation.

Importantly, despite considerable variations in mesh density, topology, and local connectivity, all generated results exhibit strong global coherence and visual consistency. There are no conspicuous artifacts or discontinuities between regions, confirming that our local-to-global generation strategy supports both creative flexibility and structural integrity across the mesh. This empowers downstream tasks—such as animation, editing, or customization—by facilitating the production of diverse, high-quality assets from unified geometric representations.

