# OpenReview forum: "MeshMosaic: Scaling Artist Mesh Generation via Local-to-Global Assembly"
_ICLR.cc/2026/Conference — ICLR 2026 Conference Withdrawn Submission_

### Official Review · Reviewer_7GEc · 2025-10-18

**Soundness:** 2
**Presentation:** 3
**Contribution:** 2
**Rating:** 4
**Confidence:** 5

**Summary:**

This paper introduces MeshMosaic, a framework to generate high-resolution triangle meshes by overcoming the sequence length limitations of prior autoregressive models. The core idea is a local-to-global assembly process: a shape is segmented into patches, and each patch is generated autoregressively, conditioned on the boundaries of its neighbors. This patch-wise approach, combined with local high-resolution quantization, enables the generation of meshes with over 100,000 triangles. The authors show that their method significantly outperforms state-of-the-art baselines in both quantitative metrics and user studies.

**Strengths:**

1. The method successfully scales autoregressive generation to much higher face counts (>100k) than previous approaches. The local quantization strategy is effective at preserving fine-grained geometric details, which is a significant practical advance.
2. The local-to-global assembly strategy, particularly the boundary conditioning mechanism (GRU-encoded boundary tokens), is a novel and well-engineered method for enforcing local continuity during patch generation.
3. The paper presents strong quantitative results across multiple datasets and metrics, consistently outperforming baselines. The user study results are also compelling, indicating a clear user preference.

**Weaknesses:**

1. MeshMosaic is built upon fine-tuning the DeepMesh model, which has known limitations in generating structurally coherent meshes for complex topologies. The proposed patch-based framework feels like an overly complicated solution designed to compensate for the weaknesses of a poor base model. The impressive polygon count masks the fact that the method's quality does not truly scale. Its performance degrades significantly on dense, complex inputs, which are the true test of scalability in today's generative landscape.
2. The paper's evaluation is misaligned with the current, most critical challenges in 3D generation. The primary industrial need is not generating meshes from clean data, but rather the high-quality re-topology of **dense, often noisy meshes produced by state-of-the-art text/image-to-3D models**. The paper completely ignores the actual SOTA in this domain (e.g., Tripo, Hunyuan3D), making its contribution appear isolated and its performance claims less impactful.
3. The local-to-global approach, even with a global frozen point cloud feature, inherently struggles with non-local properties. The authors' own example (Fig. 11) shows it fails to enforce symmetry across distant parts, a critical flaw for "artist-quality" assets where high-level structural consistency is paramount.
4. Despite efficient per-token time, the absolute wall-clock time for generating a high-resolution mesh is impractical. The reported time of "several hours to complete" for a >100K face mesh is a fatal flaw for any real-world interactive or iterative design workflow.

**Questions:**

1. A critical test for any modern mesh generation method is its ability to handle dense, AI-generated geometry. Could the authors provide a direct comparison of MeshMosaic against recent, highly relevant methods like **Tripo and Hunyuan3D's PolyGen**? The comparison should be on the task of re-topologizing dense meshes (e.g., 20K+ faces) produced by these systems, evaluating both geometric fidelity and topological quality.
2. Given the model is fine-tuned on DeepMesh, a model with known limitations, how do the authors ensure the new framework has truly fixed these weaknesses? Please provide a dedicated ablation demonstrating MeshMosaic's performance on the challenging, dense outputs mentioned in the previous question, where the original DeepMesh would likely fail completely.
3.  The global context comes from a frozen Michelangelo encoder. Why was this encoder not fine-tuned, and what evidence supports that this weak conditioning is sufficient for global coherence, especially when ablations (Fig. 13) show catastrophic failure without it?
4. The final "gluing" step suggests boundary conditioning alone is insufficient for perfect alignment. Could you quantify the magnitude of these post-hoc patch translations and discuss if they can lead to error accumulation across a mesh with dozens of patches?

---

### Official Review · Reviewer_zcRZ · 2025-10-23

**Soundness:** 3
**Presentation:** 1
**Contribution:** 3
**Rating:** 2
**Confidence:** 4

**Summary:**

MeshMosaic is a new local-to-global framework for generating high-resolution 3D artist meshes that scales beyond 100K triangles—far more than previous transformer-based models. It segments shapes into local patches generated autoregressively with shared boundaries to ensure smooth, symmetric connections. By quantizing patches separately, MeshMosaic achieves higher geometric detail, organized mesh density, and better scalability.

**Strengths:**

The general breakdown idea sounds good. It effectively reduce the computation overhead and memory bound.

**Weaknesses:**

1. The paper lacks a lot of technical details. I cannot reproduce it based on the information the paper present, especially the boundary condition part. More examples are listed in "Question".
2. Some important part are only describe in appendix vaguely. For example, the generation process inside patch is not clear. Paper only said in :871 "We developed our method on the DeepMesh codebase". Correct me if I miss it.
3. Seams is visible between different patches, like lower right in Fig13. The authors need to provide more evidence to demonstrate the overall coherence and aesthetic quality among different parts.

typo :800 Recover

**Questions:**

- :249 How to define adjacent point cloud patches?
- The order of DFS. Which adj patched to visit next, when one patch adjacent to many patches
- How the boundary triangle is encoded before GRU?
- How to define the order of boundary triangle in Transformer?
- What does the "Sliding windows" mean in Figure 6? The patch is not sliding.
- What if boundary triangles are more then 512? will it create some seams?
- When a patch is rescale and requantize to another patch, I assume some vertices will be merged into one because quantization error. Will it create topo error? how to fix?

---

### Official Review · Reviewer_kNTA · 2025-10-25

**Soundness:** 3
**Presentation:** 3
**Contribution:** 3
**Rating:** 6
**Confidence:** 4

**Summary:**

This paper introduces MeshMosaic, a novel approach for generating high-resolution and complex 3D meshes by employing a local-to-global assembly strategy. The method breaks down the complex generation task into generating smaller, local patches that are then conditioned on their boundaries to ensure seamless stitching into a cohesive, high-fidelity final mesh. This patch-based framework successfully scales the mesh complexity, demonstrating an ability to produce detailed meshes while retaining global structure and localized artistic interpretation, making it a powerful tool for generating complex 3D assets from inputs like text or point clouds.

**Strengths:**

- The paper is well written and clearly motivated.
- The patch-based generation framework is important for complex and high-resolution mesh.
- The experimental results are promising, with sound and detailed ablation study.

**Weaknesses:**

- As discussed in the limitations, the boundary conditioning is still a rather local guidance, and may fail to provide global guidance like symmetry. Also, there are cases where a part is connected to multiple boundaries, which introduces challenges to the current sequential generation setting.
- It's better to provide more failure cases to evaluate the robustness of the method.

**Questions:**

- As illustrated in Figure 5, it's possible that a patch is connected to more than 2 other previous patches. How will the boundary condition handle such cases? Will it be smoothly connected to all previous boundaries?
- Why are the boundary condition features processed by a GRU network?
- How will the number of parts affect the generation quality during inference? What's the recommended inference setting given an arbitrary point cloud or image input?

---

### Note · Authors · 2025-11-13

**Comment:**

We thank all the reviewers for their helpful suggestions and questions. Given the predominantly negative feedback from the reviewers, we have decided to withdraw this paper.

**Withdrawal Confirmation:**

I have read and agree with the venue's withdrawal policy on behalf of myself and my co-authors.